# Modulating p-AMPK/mTOR Pathway of Mitochondrial Dysfunction Caused by MTERF1 Abnormal Expression in Colorectal Cancer Cells

**DOI:** 10.3390/ijms232012354

**Published:** 2022-10-15

**Authors:** Qianqian Liu, Longlong Zhang, Yayan Zou, Ying Tao, Bing Wang, Bin Li, Ruai Liu, Boyong Wang, Lei Ding, Qinghua Cui, Jie Lin, Bingyu Mao, Wei Xiong, Min Yu

**Affiliations:** 1School of Life Sciences, Yunnan University, Kunming 650091, China; 2State Key Laboratory for Conservation and Utilization of Bio-Resources, Key Laboratory for Microbial Resources of the Ministry of Education, School of Life Sciences, Yunnan University, Kunming 650091, China; 3College of Basic Medical Sciences, Dali University, Dali 671000, China; 4Yunnan Key Laboratory of Screening and Research on Anti-Pathogenic Plant Resources from Western Yunnan, Dali University, Dali 671000, China; 5State Key Laboratory of Genetic Resources and Evolution, Kunming Institute of Zoology, Chinese Academy of Sciences, Kunming 650223, China

**Keywords:** colorectal cancer, MTERF1, cell proliferation, mtDNA, oxidative phosphorylation, AMPK/mTOR

## Abstract

Human mitochondrial transcription termination factor 1 (MTERF1) has been demonstrated to play an important role in mitochondrial gene expression regulation. However, the molecular mechanism of MTERF1 in colorectal cancer (CRC) remains largely unknown. Here, we found that MTERF1 expression was significantly increased in colon cancer tissues compared with normal colorectal tissue by Western blotting, immunohistochemistry, and tissue microarrays (TMA). Overexpression of MTERF1 in the HT29 cell promoted cell proliferation, migration, invasion, and xenograft tumor formation, whereas knockdown of MTERF1 in HCT116 cells appeared to be the opposite phenotype to HT29 cells. Furthermore, MTERF1 can increase mitochondrial DNA (mtDNA) replication, transcription, and protein synthesis in colorectal cancer cells; increase ATP levels, the mitochondrial crista density, mitochondrial membrane potential, and oxygen consumption rate (OCR); and reduce the ROS production in colorectal cancer cells, thereby enhancing mitochondrial oxidative phosphorylation (OXPHOS) activity. Mechanistically, we revealed that MTERF1 regulates the AMPK/mTOR signaling pathway in cancerous cell lines, and we also confirmed the involvement of the AMPK/mTOR signaling pathway in both xenograft tumor tissues and colorectal cancer tissues. In summary, our data reveal an oncogenic role of MTERF1 in CRC progression, indicating that MTERF1 may represent a new therapeutic target in the future.

## 1. Introduction

Colorectal cancer (CRC) is the third most malignant tumor and one of the leading causes of cancer-related death in the world [1]. At present, the incidence of CRC among adults under the age of 50 is particularly severe, with an annual growth rate of 2.2% since 1995. The five-year relative survival rate of CRC patients is 65%, which dropped to 12% in stage IV CRC patients [2]. Therapy is usually less extensive and more successful when CRC is detected at an early stage, while approximately 15–25% of all CRC patients have distant metastases (TNM Stage IV) at the time of the primary [3,4]. Furthermore, CRC cells develop resistance to chemotherapy and radiation due to their intratumor heterogeneity and clonal evolution; thus, combined-modality therapy has failed to universally improve patient prognosis [5]. Therefore, the molecular mechanisms must be further explored, and more effective therapeutic target genes must be developed.

Mitochondria are complex organelles that influence cancer initiation, growth, survival, and metastasis, and many facets of mitochondrial biology, including mitochondrial mass, dynamics, cell death regulation, redox homeostasis, metabolic regulation, and signaling actively contribute to tumorigenesis [6]. Mitochondria are biological powerhouses that provide cellular fuel through the conversion of nutrients to energy through OXPHOS [7]. OXPHOS is upregulated in different tumors, such as lymphomas, leukemias, melanomas, breast, pancreatic, lung, and endometrial cancers, and OXPHOS upregulation is consistent with the increased mitochondrial activity related to tumorigenesis as well as to anchor-age-independent growth and resistance to chemotherapy [8]. Importantly, the processes of energy production via OXPHOS are directly or indirectly linked to mitochondrial gene expression, which is the process by which mtDNA is transcribed into RNA and subsequently translated into protein [9]. Inhibition of mtDNA gene expression by treatment with the small-molecule inhibitors IMTs significantly downregulated OXPHOS activity in ovarian carcinoma cells [10]. MtDNA is composed of heavy and light strands (H-strand and L-strand, respectively), and the H-strand has more purine nucleotides than the L-strand. Mitochondrial transcription termination factor (MTERF) family members (MTERF1, 2, 3, and 4) play an important role in mitochondrial gene expression [11,12]. MTERF1 is an mtDNA binding protein that consists of 342 amino acids and modulates the expression of mitochondrial genes by preventing L-strand transcription interference within mtDNA [13,14,15]. In addition, MTERF1 has been shown to stimulate more general transcription pausing and arrest mtDNA replication [16,17]. MTERF1 has multiple binding sites in the human mitochondrial genome and displays extreme flexibility to accommodate a wide range of interactions with mtDNA [9,16]. Therefore, the role of MTERF1 is more complex than previously thought.

MTERF1 plays an important role in the occurrence and development of tumors. Overexpression of MTERF1 increased mitochondrial gene transcription and promoted ATP synthesis and cell proliferation in HeLa cells [18]. The mRNA level of MTERF1 is significantly increased in non-small cell lung cancer (NSCLC), and it is closely related to the improvement of the overall survival (OS) rate in patients with lung adenocarcinoma [19]. Moreover, MTERF1 can regulate mitochondrial activity and cell proliferation by mediating the gene expression of mtDNA [17,19]. Recent studies have also shown that MTERF1 causes gene amplification in a variety of solid tumors, and its abnormal protein expression may be involved in the development and metastasis of malignant tumors [19]. These studies implied that MTERF1 may be an oncogene. However, its mechanism in tumorigenesis is still unclear.

Our work confirmed that MTERF1 was highly expressed in CRC tissues and CRC cells and showed that MTERF1 promoted CRC progression by increasing mitochondrial OXPHOS and regulating the p-AMPK/mTOR signaling pathway.

## 2. Results

### 2.1. MTERF1 Is Highly Expressed in CRC Tissues

To investigate the expression of MTERF1 in CRC, we first performed Western blotting and immunohistochemistry (IHC) analyses to evaluate the protein expression profiles of MTERF1 between cancer and normal tissues. The results showed that the expression of MTERF1 in CRC tissues was significantly higher than that in adjacent cancers (Figure 1A–C). Then, we validated that the protein expression of MTERF1 in the CRC tissue microarray (TMA) was consistent with the results of Western blotting and IHC analysis (Figure 1D). Statistical analysis suggested that a high expression level of MTERF1 protein in CRC patients was strongly correlated with four clinicopathological parameters of CRC, namely, patient lymph node status (*p* = 0.0044, χ^2^ = 8.1251), tumor stage (*p* = 0.0019, χ^2^ = 9.6651), nodal stage (*p* = 0.0044, χ^2^ = 8.1251), and clinical stage (*p* = 0.0044, χ^2^ = 8.1251) (Appendix A). Thus, MTERF1 evaluations could help clinicians improve prognosis prediction and make better therapeutic decisions. However, a high expression level of MTERF1 protein was not correlated with patient age or sex, hypertension, diabetes, tumor size, ER expression, or PR expression in CRC (*p* > 0.05) (Appendix A). These results indicate that MTERF1 is significantly upregulated in CRC tissues and may play an important role in the occurrence and development of CRC.

### 2.2. MTERF1 Promotes the Proliferation, Migration, and Invasion of CRC Cells and Affects Cell Cycle Transition and Apoptosis

To further explore the role of MTERF1 in CRC and the mechanisms of its oncogenic function, we first analyzed the relative expression level of MTERF1 protein in a normal human colorectal cell line (FHC) and CRC cancerous cell lines (HT29, HCT8, SW480, Caco2, HCT116, and RKO). The data revealed that the expression of MTERF1 was upregulated in all CRC cancerous cell lines except for HT29 (Figure 2A). We then selected two CRC cancerous cell lines with different expression levels (HCT116 cells with relatively higher expression levels of MTERF1 and HT29 cells with relatively lower expression) to further explore the function of MTERF1. We overexpressed and inhibited the expression of MTERF1 using lentiviral packaging (see Materials and Methods) and verified the overexpression and inhibition efficiency by comparing both relative protein and mRNA expression levels in the cell lines targeted by overexpression and knockdown of MTERF1 (Figure 2B,C). We further tested whether MTERF1 regulates CRC cell proliferation by growth curve, CCK-8, and cell colony formation assays. The data showed that overexpressing MTERF1 promoted cell proliferation while knockdown of MTERF1 in CRC distinctly inhibited cell proliferation (Figure 2D–F).

Next, we tested whether MTERF1 regulates CRC cell migration and invasion through wound healing and transwell assays. Our data showed that MTERF1 overexpression in HT29 cells indeed markedly increased cell migration and invasion, while MTERF1 depletion in HCT116 cells dramatically decreased cell migration and invasion (Figure 2G,H). Based on these results, we hypothesized that MTERF1 might be important for CRC cell cycle regulation. To test this hypothesis, flow cytometry (FACS) was performed to measure the cell cycle of CRC cells. Our data showed that MTERF1 overexpression significantly reduced the G1 phase ratio and increased the S phase ratio (Appendix A). The protein level of cyclin D1 (G1/S phase checkpoint protein) was upregulated in HT29 cells overexpressing MTERF1, whereas that of p21 (cyclin-dependent kinase inhibitor) was decreased and cyclin B1 (G2/M phase checkpoint protein) remained unchanged, respectively (Appendix A). To determine the effect of MTERF1 on the survival of CRC cells, cell apoptosis was evaluated by flow cytometry. The results showed that MTERF1 overexpression significantly reduced the proportion of apoptotic CRC cells (Appendix A). The cleavage and activation of caspase-3, caspase-8, and nuclear poly (ADP-ribose) polymerase (PARP) have been confirmed as critical steps in the apoptosis signaling pathway. We then detected the expression levels of caspase-8 and cleaved and activated both caspase-3 and PARP using Western blotting. The data showed that caspase-8 was strongly downregulated upon MTERF1 overexpression in HT29 cells, the levels of full-length caspase-3 and PARP were strengthened, but cleaved caspase-3 and PARP were dramatically reduced (Appendix A). Importantly, in the loss-of-function assays, the results showed that the knockdown of MTERF1 significantly increased the G1-phase ratio, decreased the S-phase ratio, and induced apoptosis in HCT116 cells, which was consistent with the expression levels of cell cycle and apoptotic proteins (Appendix A). These results demonstrated that MTERF1 promotes CRC cell cycle transition at the G1/S phase and inhibits CRC apoptosis.

### 2.3. MTERF1 Promotes CRC Xenograft Tumor Formation

To further assess whether MTERF1 is important for CRC progression in vivo, we performed a tumorigenesis assay in nude mice. Twenty-five male nude mice at five weeks of age were randomly divided into five groups and then injected with HT29 and HCT116 cells stably expressing pCDH, pCDH-MTERF1, scramble control shRNA, or two MTERF1-targeting shRNAs (5 × 10^5^ cells/point subcutaneously). Mice were monitored each day and sacrificed after 28 days, and tumors were harvested and weighed. Our results showed that the tumor sizes in the MTERF1 overexpression group were significantly larger than those in the control group (Figure 3A–C). Knockdown of MTERF1 in HCT116 cells dramatically retarded tumor weights and volumes compared to scrambled shRNA-transformed cells (Figure 3A–C). Subsequently, we extracted total proteins from the xenograft tumor and verified the overexpression and inhibition efficiency of MTERF1 in xenograft tumors (Figure 3D). In addition, IHC staining for Ki67 in xenograft tumor tissues showed that MTERF1 indeed promoted cancerous cell proliferation in vivo (Figure 3E). In summary, MTERF1 expression promoted tumor formation in vivo.

### 2.4. MTERF1 Positively Regulates Mitochondrial Gene Replication, Transcription, and OXPHOS

To investigate whether MTERF1 regulated the replication of CRC cellular mtDNA, the noncoding D-loop region of mtDNA was amplified by quantitative real-time PCR (qPCR) and normalized to the housekeeping gene 18S rDNA. We found an obvious increase in the mtDNA copy number after MTERF1 overexpression (Figure 4A). In contrast, the copy number of mtDNA decreased by 0.7 times (MTERF1-RNAi#1) and 0.4 times (MTERF1-RNAi#2) compared with the control cells (RNAi -vector) after MTERF1 knockdown (Figure 4A). These data indicated that the copy number of mtDNA was regulated by MTERF1. The process by which mtDNA is transcribed into RNA and subsequently translated into protein is vital to OXPHOS. To determine the effect of MTERF1 on mtDNA transcription, we selected five representative mitochondrial genome genes, namely, 12S rRNA, 16S rRNA, mitochondrial NADH-ubiquinone oxidoreductase chain 1 (ND1), mitochondrial cytochrome b (Cytb) in the H-stand and NADH dehydrogenase subunit 6 (ND6) in the L-stand, and two nuclear genome-coded genes, namely, mitochondrial transcription factor A (TFAM) and NADH: ubiquinone oxidoreductase subunit B8 (NDUFB8), to perform qRT–PCR experiments. The results showed that the transcription levels of 12S rRNA, 16S rRNA, ND1, Cytb, ND6, and TFAM were significantly upregulated with MTERF1 overexpression and significantly downregulated with MTERF1 knockdown while the transcription level of NDUFB8 remained unchanged with both MTERF1-overexpressing and MTERF1-knockdown CRC cells (Figure 4B). In addition, the changes in ND1 and TFAM protein levels were consistent with the changes in their mRNA levels (Figure 4C). We also found that ND1 and TFAM protein levels were higher in the CRC tissues of patients than in paired adjacent noncancerous tissues and normal tissues (Figure 4D).

The above experiments confirmed that MTERF1 increased the mtDNA copy number and promoted the expression of mitochondrial genes. Interestingly, the intracellular ATP levels were 2.4 times higher in MTERF1-overexpressing cells than in control cells, and ATP production was significantly reduced in MTERF1 knockdown cells (Figure 4E), indicating that MTERF1 promoted the production of mitochondrial ATP in CRC cells. The reactions of OXPHOS generated the mitochondrial membrane potential (ΔΨm) and drove the majority of ATP production in respiring cells. Therefore, we speculated that MTERF1 may regulate mitochondrial OXPHOS activity. FACS was used to detect the effect of MTERF1 on the ΔΨm of CRC cells, and our data showed that the ΔΨm was significantly increased after MTERF1 was overexpressed and significantly decreased in MTERF1 knockdown cells (Figure 4F). To further confirm the impact of MTERF1 on OXPHOS, we measured the oxygen consumption rate (OCR) of colorectal cancer cells. The results showed that the OCR increased in HT29 cells when MTERF1 was overexpressed and decreased when MTERF1 was knocked down (Figure 5A). We also detected the expression level of ROS and superoxidation by flow cytometry (Figure 5B). The quantitative results showed that the ROS levels were decreased in MTERF1-overexpressing cells, and the ROS and superoxidation levels were significantly increased in MTERF1 knockdown cells (Figure 5C,D). Together, these results indicate that MTERF1 promotes mitochondrial gene replication, transcription, and translation and improves mitochondrial OXPHOS activity in CRC cells.

### 2.5. MTERF1 Expression Changes the Mitochondrial Inner Membrane Ultrastructure

The above results have shown that MTERF1 plays an important role in mitochondrial function, which is closely linked to organelle morphology and structure. Therefore, we examined the mitochondrial morphology following the upregulation and downregulation of MTERF1 via transmission electron microscopy analysis. The results revealed that the knockdown of MTERF1 in HCT116 cells not only reduced the percentage of mitochondria with lamellar cristae but also clearly reduced the crista density and length (Figure 5E–J). In addition, overexpression of MTERF1 improved the crista density in HT29 cells (Figure 5G). Mitochondrial cristae are folds within the inner mitochondrial membrane, and an increase in crista density can lead to increased rates of respiration [20]. Therefore, our results indicate that MTERF1 is essential for maintaining the mitochondrial inner membrane ultrastructure and is involved in OXPHOS activity in CRC cells.

### 2.6. MTERF1 Expression Correlates with AMPK/mTOR Pathway Activation in Human CRC Cells

Our data demonstrated that MTERF1 promoted an increase in ATP production. ATP not only supports energy storage within cells but is also a transmitter/signaling molecule that serves intercellular communication [21]. AMP-activated protein kinase (AMPK) is an important energy sensor in cells, and AMPK is activated due to a decrease in intracellular ATP levels [22]. The AMPK-mTOR pathway plays an important role in regulating tumor cell proliferation. We collected MTERF1-overexpressing or MTERF1-knockdown cells to detect the expression of proteins involved in the AMPK-mTOR pathway. The results showed that compared with control cells, AMPK expression levels did not change significantly, while phosphorylated AMPK (P-AMPK) levels were significantly reduced in HT29 cells with MTERF1 overexpression. Significant changes were not observed in the target of rapamycin (mTOR), although the protein levels of p-mTOR, p-P70S6K (P70S6K: ribosomal protein S6 kinase), and p-4E-BP1 (4E-BP1: eukaryotic initiation factor 4E-binding protein 1) were increased (Figure 6A). These protein levels showed an opposite trend compared with HCT116 cells with MTERF1 knockdown (Figure 6A), suggesting that MTERF1 may promote the occurrence and development of CRC cells by regulating the AMPK/mTOR signaling pathway.

To further verify our experimental results, we blocked the AMPK/mTOR signaling pathway by using the mitochondrial ATPase inhibitor oligomycin A and mTOR inhibitor rapamycin. The results showed that the AMPK/mTOR pathway regulated by MTERF1 was blocked by oligomycin and rapamycin (Figure 6B). We added 5 μM oligomycin A and 300 nM rapamycin to HT29 cells to detect ATP levels and cell proliferation. The results showed that the ATP level and cell number of HT29 cells decreased significantly after treatment with oligomycin A and rapamycin, and the ATP level and cell number dropped sharply in MTERF1-overexpressing cells (Figure 7A,B). In addition, we also detected the protein expression levels of the AMPK/mTOR signaling pathway in xenograft tumor tissues and clinical tumor tissues, and the results were consistent with previous results in cancer cells. Compared with the adjacent tissues, MTERF1 was highly expressed, p-AMPK was suppressed, p-mTOR was increased, and the downstream effector proteins p-mTOR, P70S6K, and 4E-BP1 were phosphorylated in tumor tissues (Figure 7C,D).

In summary, our data indicated that MTERF1 increases mitochondrial OXPHOS and intracellular ATP levels by regulating the replication, transcription, and translation of mtDNA. The increase in ATP content inhibited p-AMPK, which in turn activated the phosphorylation level of mTOR and its major downstream effectors 4EBP1 and P70S6K and ultimately promoted the proliferation, migration, and invasion of CRC cells (Figure 7E).

## 3. Discussion

In this study, we first confirmed that MTERF1 was highly expressed in CRC cancerous tissues compared with normal, noncancerous tissues. Then, we examined the protein expression of MTERF1 in CRC cancerous cell lines compared with a colon epithelial cell line and found that MTERF1 was also upregulated in CRC cancerous cells. Through overexpression of MTERF1 in the HT29 cell line and depletion of MTERF1 in the HCT116 cell line, we provided evidence that MTERF1 promotes CRC cell proliferation, migration, and invasion in vitro promotes cell cycle progression at the G1/S phase, inhibits cellular apoptosis, and promotes xenograft tumor formation in vivo. These results indicate that MTERF1 functions as an oncogenic factor to facilitate CRC progression.

In mammals, mitochondria play an important role in OXPHOS, the tricarboxylic acid cycle (TCA), intracellular calcium signal transduction, cell apoptosis, and reactive oxygen generation [23,24]. Most importantly, mitochondria produce a large amount of ATP via OXPHOS to maintain the energy required for cell growth [25]. ATP is an energy indicator of cells, and an increase in ATP levels in cells can promote cancer cell EMT, migration, and invasion [26]. It is worth noting that all mitochondrial functions are directly or indirectly related to OXPHOS, and the regulation of mtDNA replication and transcription is crucial to mitochondrial OXPHOS [10]. MTERF1 is a key regulator of mammalian mtDNA transcription and replication [9,16], and it plays a significant role in the regulation of cancer cell proliferation [17,18,19]. Here, we found that MTERF1 promoted mtDNA replication, transcription and improved the mitochondrial crista density, mitochondrial membrane potential, mitochondrial ATP production, oxygen consumption rate (OCR), and ROS production in colorectal cancer cells. Our results provide mechanistic insights into MTERF1 regulation in the proliferation, migration and invasion of CRC cells by regulating the replication, transcription and translation of mtDNA and enhancing the activity of mitochondrial OXPHOS.

The mechanism underlying the ability of MTERF1 to positively regulate mtDNA replication and gene expression in CRC remains unknown. Previous evidence has highlighted the role of MTERF1 proteins in the control of mitochondrial gene expression and suggested that MTERF1 can mediate the formation of a transcriptional loop between the heavy strand promoter (HSP) and termination site that facilitates rRNA synthesis [27]. However, that model was not supported by further studies, which showed that MTERF1 binds mtDNA to prevent transcriptional interference at the L-strand promoter but is dispensable for rRNA gene transcription regulation in mice [15] and that MTERF1 did not play a role in the termination of HSP-driven transcription in HEK293 cells [28]. In addition, MTERF1 has been shown to stimulate more general mtDNA transcription pausing as well as replication pausing to avoid mtDNA damage [16,17]. In CRC cells, MTERF1 promoted mtDNA replication and transcription. The expression of mitochondrial transcription factor A (TFAM) was regulated upon MTERF1 overexpression and knockdown, and in CRC tissues, the higher expression level of TFAM was similar to that of MTERF1. A previous study reported that TFAM is essential for human mtDNA transcription and that the expression of TFAM results in the upregulation of the mtDNA copy number [29]. MTERF1 may regulate mtDNA replication and transcription through a mechanism that might involve additional proteins, and this important point requires further investigation.

AMPK and mTOR are the key regulators of cellular energy in cancers. The findings by Zhang et al. implied that polysaccharide-induced mitochondrial dysfunction and cytotoxic autophagy repressed the propagation of colon cancer cells via ROS-ATP-AMPK signaling, thus providing a new opinion for the study of antineoplastic drugs [30]. Liu et al. demonstrated that sorafenib kills liver cancer cells by disrupting SCD1-mediated synthesis of monounsaturated fatty acids via the ATP-AMPK-mTOR-SREBP1 signaling pathway [31]. AMPK is activated when ATP is depleted, and AMP levels are increased, thereby directly inhibiting multiple targets, including mTORC1, to restore energy homeostasis. AMPK activation can overcome multidrug resistance by decreasing the ATP-binding cassette (ABC) transporters (drug efflux pumps) that cause the efflux of hydrophobic compounds and xenobiotics such as chemotherapeutic drugs [32,33]. ABC transporter depends on the hydrolysis of ATP, and low ATP concentration has the strong ability to inhibit the ABC transporters [34]. The inhibition of mTOR also increases the drug sensitivity of cancer cells [35]. Therefore, targeting AMPK/mTOR can effectively inhibit tumor formation and enhance chemotherapeutic sensitivity, and disruption of ATP level may be a more straightforward strategy to overcome drug resistance [36]. Intriguingly, we found that MTERF1 knockdown contributed to the decrease in ATP levels in CRC cells, thereby activating p-AMPK and then inhibiting p-mTOR and the downstream effector proteins of p-mTOR, p-P70S6K, and p-4E-BP1, indicating that MTERF1 regulates tumor cell proliferation by the AMPK/mTOR signaling axis.

In summary, this work has provided strong evidence indicating that MTERF1 is upregulated in CRC tissues and that the oncogenic role of MTERF1 in CRC cell proliferation and tumorigenesis is achieved by enhancing mitochondrial OXPHOS activity and regulating the AMPK/mTOR pathway. Thus, MTERF1 might act as a promising target for the diagnosis and therapeutic strategy of CRC.

## 4. Materials and Methods

### 4.1. Clinical Samples and Data

A total of 21 pairs of primary tumor tissues and 22 corresponding normal tissues used for IHC assays were collected from CRC patients who received surgical treatment at the Department of Pathology of Dali People’s Hospital (Dali, China) between May and August 2018. The CRC cancerous tissues and paired adjacent noncancerous tissues used for Western blotting assays were obtained from 15 cases at the Department of Colorectal Surgery, The Third Affiliated Hospital of Kunming Medical University, from March and May 2018. Tissue microarray (TMA) analysis of 41 normal tissues and 52 cancer samples was provided by Wuhan Seville Biotechnology Co., Ltd. (Wuhan, China). The patients signed informed consent forms, and the work was approved by the Ethics Committee of Yunnan University (CHSRE2021013).

### 4.2. Cell Culture

Human embryonic kidney 293T cells and human normal colorectal FHC cells were provided by the Laboratory of Biochemistry and Molecular Biology of Yunnan University (Kunming, China). Human colon cancer cell lines were purchased from the Kunming Cell Bank of the Chinese Academy of Science. HT29, HCT8, SW480, Caco2, HCT116, and RKO cells were all cultured in DMEM (Gibco, Carlsbad, CA, USA) supplemented with 10% fetal bovine serum (BI, Kibbutz Beit Haemek, Israel) and 1% penicillin/streptomycin (Gibco), and FHC cells were cultivated in RPMI-1640 medium (Gibco). All cells were cultured at 37 °C in a 5% CO_2_ humidified environment. Mycoplasma was routinely tested using a myco-plasma detection kit (TransGen Biotech, Beijing, China). Oligomycin A and rapamycin used for cell treatment were obtained from MedChem Express (Shanghai, China).

### 4.3. Plasmids, Transfection, and Infection

The human MTERF1 gene was PCR-amplified from cDNA and cloned into a pCDH-CMV-MCS-EF1-GFP lentiviral vector (MTERF1-pCDH). To silence MTERF1, two human MTERF1-targeting short hairpin RNA (shRNA) sequences were cloned into a pLKO.1-TRC vector to generate the respective MTERF1-RNAi(s). Independent shRNA targeting sequences were MTERF1-shRNA#1, 5′-AACACGTACTCCCGAG AATCT-3′ and MTERF1-shRNA#2, 5′-AGTCT TGATCTGAATAAACAG-3′. The control scramble shRNA (RNAi-Vector) sequence was 5′-GCACTACCAGAGCTAACTCAG-3′. The newly built constructs were transfected into 293T cells to produce a lentivirus. The cells were infected by viruses twice for 48 and 72 h with viral supernatants containing 4 μg/mL polybrene and then selected with 3 μg/mL puromycin (Solarbio, Beijing, China) to obtain stably infected cell lines.

### 4.4. RNA Extraction, DNA Extraction, and Quantitative Real-Time PCR (qPCR)

Total RNA in the tissues and cells was extracted with TRIzol reagent (Solarbio), and first-strand cDNA was inversely transcribed from total RNA (2 μg) using M–MLV reverse transcriptase (Promega, Madison, WI, USA). Then, RT–qPCR was performed to analyze cDNA using SYBR Green (Takara, Tokyo, Japan) on an ABI Prism 7500 Sequence Detection System (Applied Biosystems, USA), and the data were normalized to GAPDH mRNA levels. The expression levels of mRNA were calculated using the comparative CT (2-∆∆CT), and all experiments were performed with three biological replicates. PCR was performed under the following conditions: 95 °C for 5 min, followed by 40 cycles of 95 °C for 30 s and 60 °C for 40 s. For the mtDNA replication assay, DNA was isolated from CRC cells using a TIANamp Genomic DNA Extraction Kit (Tiangen, Beijing, China) according to the manufacturer’s protocols. The expression level of the D-loop of mtDNA was detected to represent the mtDNA replication level, and 18S rDNA was used as an internal reference. The primer sequences used in qPCR are listed in Appendix A.

### 4.5. Immunohistochemical (IHC) and Immunofluorescence (IF) Staining

IHC was performed as previously described (Liu et al., 2020). The primary antibodies were as follows: anti-MTERF1 (1:250, Abcam, Cambridge, MA, USA) and anti-Ki67 (1:200, Abcam, USA). The staining index (SI) was calculated as the tissue staining intensity and percentage. The percentage of cells was graded as follows: 1 (0–25%), 2 (26–50%), 3 (51–75%), or 4 (>75%); staining intensity was scored as follows: 0 (negative), 1 (weak), 2 (moderate), and 3 (strong); and the percentage of cells scored in the four categories was as follows: 1 (0–25%), 2 (26–50%), 3 (51–75%), or 4 (>75%). SI = staining percentage × intensity. Samples with SI ≥ 6 were determined to have high expression and those with SI < 6 were determined to have low expression.

### 4.6. Cell Proliferation, Viability, and Colony Formation Assays

For the cell proliferation assays, the tested cell lines were plated on 24-well plates, and the cell numbers were subsequently counted each day. For the cell viability assays, the cells were plated at a density of 5000 cells/well in 96-well plates and incubated overnight. The cells were then treated with CCK-8 at 0, 24, 48, and 72 h and assessed using a microplate reader at 450 nm. For the cell colony formation, the cells were seeded at a density of 20,000 cells/well in 6-well plates (three wells per group). The cells in 6-well plates were cultured for 7–18 days, with the medium changed every 2–3 days. To observe the results, the cells were fixed in 4% paraformaldehyde for 30 min and stained with purple crystal solution (Sigma-Aldrich, St. Louis, MO, USA) for 30 min.

### 4.7. Wound Healing and Invasion Assay

The migration assay was performed as previously described (Hu et al., 2018). Invasion assays were performed using transwell chambers (Corning, New York, NY, USA). One hundred microliters of Matrigel matrix (Corning) were added to the upper chamber and incubated at 37 °C for 1 h for the invasion assay. Two hundred microliters of a cell suspension of approximately 3 × 105 were seeded in the upper chambers with serum-free DMEM, while the lower chamber was filled with 10% FBS. Cells were stained with crystal violet (Beyotime) and photographed by microscopy (Olympus, Tokyo, Japan).

### 4.8. Xenograft Tumor Formation Assay

A total of 25 male nude mice at 4 weeks of age were purchased from the SPF Animal Experiment Center of Kunming Medical University (Kunming, China) and randomly divided into the indicated groups. Five BALB/c nude mice in each group were injected subcutaneously with HT29 cells stably overexpressing MTERF1 and HCT116 cells with knockdown of MTERF1 (5 × 10^5^ cells/point subcutaneously), and all mice were sacrificed 28 days later. Tumors were then harvested and weighed. A portion of the tumor tissues was fixed in 4% paroformaldehyde and embedded in paraffin for IHC and IF analysis. Animal experiments were approved by the Animal Ethics Committee of Yunnan University (YNUCARE20210074).

### 4.9. Western Blotting Analysis

Total proteins from the cells or tissues were extracted by RIPA buffer with protease inhibitors (Beyotime Institute of Biotechnology, China) on ice. Supernatants were boiled at 100 °C for 7 min, and total protein samples (20 µg) were separated by electrophoresis and transferred to polyvinylidene fluoride membranes (Millipore, Bedford, MA, USA). The following primary antibodies were used: anti-MTERF1 (Abcam, 1:1000; USA), anti-ND1 (Anbo, 1:1000; Changzhou, China), anti-TFAM (Abcam, 1:1000), anti-p21 (Abcam, 1:500), anti-cyclinB1 (Cell Signaling Technology, 1:1000; Danvers, MA, USA), anti-cyclinD1 (Abcam, 1:1000), anti-caspase-8 (Cell Signaling Technology, 1:1000), anti-caspase-3 (Cell Signaling Technology, 1:1000), anti-PARP (Cell Signaling Technology, 1:1000), anti-AMPKα-1 (Abcam, 1:1000), anti-p-AMPKα-1 (Abcam, 1:1000), anti-mTOR (Cell Signaling Technology, 1:1000), anti-p-mTOR (Cell Signaling Technology, 1:1000), anti-P70S6K (Cell Signaling Technology, 1:1000), anti-p-P70S6K (Cell Signaling Technology, 1:1000), anti-4E-BP1 (Cell Signaling Technology, 1:1000) and anti-p-4E-BP1 (Cell Signaling Technology, 1:1000). To confirm equal sample loading, the membranes were stripped and reprobed using an-ti-α-tubulin antibody (Beyotime, 1:1000, Nantong, China) and anti-GAPDH antibody (Beyotime, 1:1000).

### 4.10. Flow Cytometric Analyses of Cell Cycle and Apoptosis

For the cell cycle analysis, harvested cells were washed twice in PBS and fixed in ice-cold 75% ethanol for 48 h. The cells were then washed twice with cold PBS and incubated in PBS mixed with 50 μg/mL propidium iodide (Sigma) and 20 μg/mL RNase A for 30 min at 37 °C. The samples were run on a flow cytometer (BD, LSR Fortessa, San Jose, CA, USA), and the data were analyzed using FlowJo X software (BD, LSR Fortessa, San Jose, CA, USA). For the analysis of HT-29 stable cell line apoptosis, the Annex-in V-PE/7-AAD detection kit (Abnova, Taipei, China) was used according to the manufacturer’s instructions. For the analysis of HCT116 stable cell line apoptosis, apoptosis was measured using the Annexin V-FITC/PI detection kit (BD, USA) according to the manufacturer’s instructions. The data were analyzed using FlowJo X software.

### 4.11. Mitochondrial Membrane Potential Measurement

Dissipation of the mitochondrial membrane potential (MMP) is a hallmark of mitochondrial function. The cationic dye JC-1 stains the mitochondria of healthy cells red and unhealthy cells green. JC-1 (200×) stock solution was diluted in DMEM and vortexed. A total of 8 × 105 cells per well were added to the JC-1/DMEM medium and incubated for 20 min in the dark at 37 °C. The cells were harvested in 1× PBS and then analyzed for red and green fluorescence by flow cytometry, and the data were analyzed using FlowJo X software.

### 4.12. Determination of ATP Levels

Cellular ATP levels were measured using a luciferin/luciferase assay. The cells were permeabilized prior to the addition of luciferin substrate and luciferase provided in the ATP assay kit (Beyotime, Shanghai, China). Bioluminescence was assessed on a LUMAT LB9507 (EG&G Berthold, Bad Wildbad, Germany). The diluted ATP standard solution was measured with a luminometer, the relative light units (RLUs) were measured with a Pil02 ATP fluorometer (Hygiena, Camarillo, CA, USA), and a standard curve was calculated for the ATP concentration. The ATP content was normalized by the protein content.

### 4.13. ROS/Superoxide Detection Assay

A ROS/Superoxide Detection Assay Kit (Abcam) was used to directly monitor the real-time production of reactive oxygen species (ROS) in cancer cells. Briefly, cells were harvested from PDA in a 1.5 mL centrifuge tube, treated with 500 µL of ROS/Superoxide Detection Solution, and then incubated for 60 min at 37 °C in the dark. The stained cells were analyzed by flow cytometry, and the data were analyzed using FlowJo X software. Oxidative Stress Detection Reagent (Green, Ex/Em 490/525 nm) was used to evaluate the total ROS, and Superoxide Detection Reagent (Orange, Ex/Em 550/620 nm) was used to evaluate superoxide.

### 4.14. OCR (Rates of Oxygen consumption) Detection Assay

Intact cells were permeabilized using a Mitochondrial Stress Test Complete Assay Kit (Abcam) to detect the OCR according to the manufacturer’s instructions. The extracellular O_2_ probe is quenched by O_2_ through molecular collision; thus, the amount of fluorescence signal is inversely proportional to the amount of extracellular O_2_ in the sample. After baseline oxygenation was established, the OCRs were calculated from the changes in the fluorescence signal over time.

### 4.15. Transmission Electron Microscopy

The harvested cells were fixed in ice-cold 2% glutaraldehyde in 0.1 M sodium cacodylate buffer (pH = 7.4) at 4 °C. The cells were then rinsed with sodium cacodylate buffer and further fixed in 1% OsO_4_ at 4 °C for 2 h before dehydration with acetone. The cell pellets were embedded in resin before polymerization at 60 °C for 48 h. Ultrathin sections (60 nm) were mounted onto copper grids and counterstained with 2% uranyl acetate and lead citrate before observation under a transmission electron microscope (JEM 1400 plus, Japan) operating at 120 kV.

### 4.16. Statistical Analyses

All quantitative data are presented as the mean ± standard error of the mean (SEM) from at least three independent experiments. Statistical analyses were carried out using ImageJ, SPSS version 20.0, and GraphPad Prism 7.0 (GraphPad Software, Inc., San Diego, CA, USA). The experiments were analyzed using an unpaired Student’s *t*-test for two groups. One-way analysis of variance (ANOVA) was used for comparisons of more than two groups. A value of *p* < 0.05 was considered statistically significant.

## Figures and Tables

**Figure 1 ijms-23-12354-f001:**
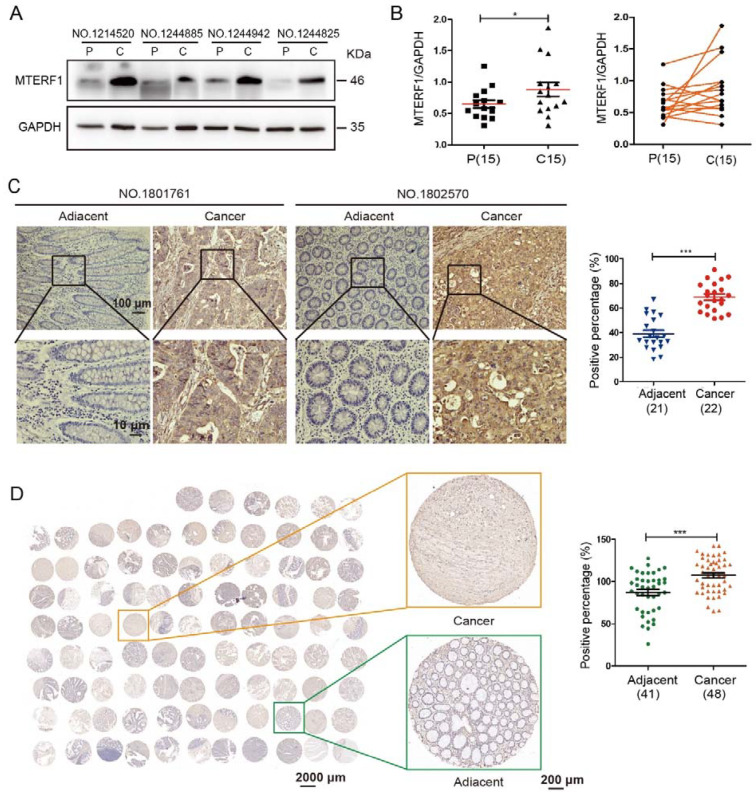
Expression analysis of MTERF1 protein in clinical tissues of CRC. (**A**,**B**) Western blotting analysis (**A**) and quantification (**B**) of MTERF1 expression in CRC tissues and paired adjacent normal tissues. P, paired adjacent normal tissues; C, CRC tissues. The diagonal line (**B**) connects the CRC tissues with paired adjacent normal tissues. GAPDH was used as an internal control. (**C**) Representative image of IHC staining of the expression of MTERF1 protein in clinical tumor tissues from CRC patients and corresponding adjacent tissues. Quantification of IHC is shown on the right. Scale bar: 100 μm. (**D**) Colorectal cancer TMAs that contain colon cancer tissues and paired adjacent normal tissues were stained with MTERF1 antibodies, and the quantification data for IHC are shown. Representative images with differential magnification of immunohistochemistry (IHC) staining are shown. Orange and green squares indicate higher magnification views of left regions boxed. Scale bar: 2000 μm. * *p* <0.05, *** *p* <0.001 by two-tailed Student’s *t*-test. Data represent the mean ± SD from three independent biological replicates.

**Figure 2 ijms-23-12354-f002:**
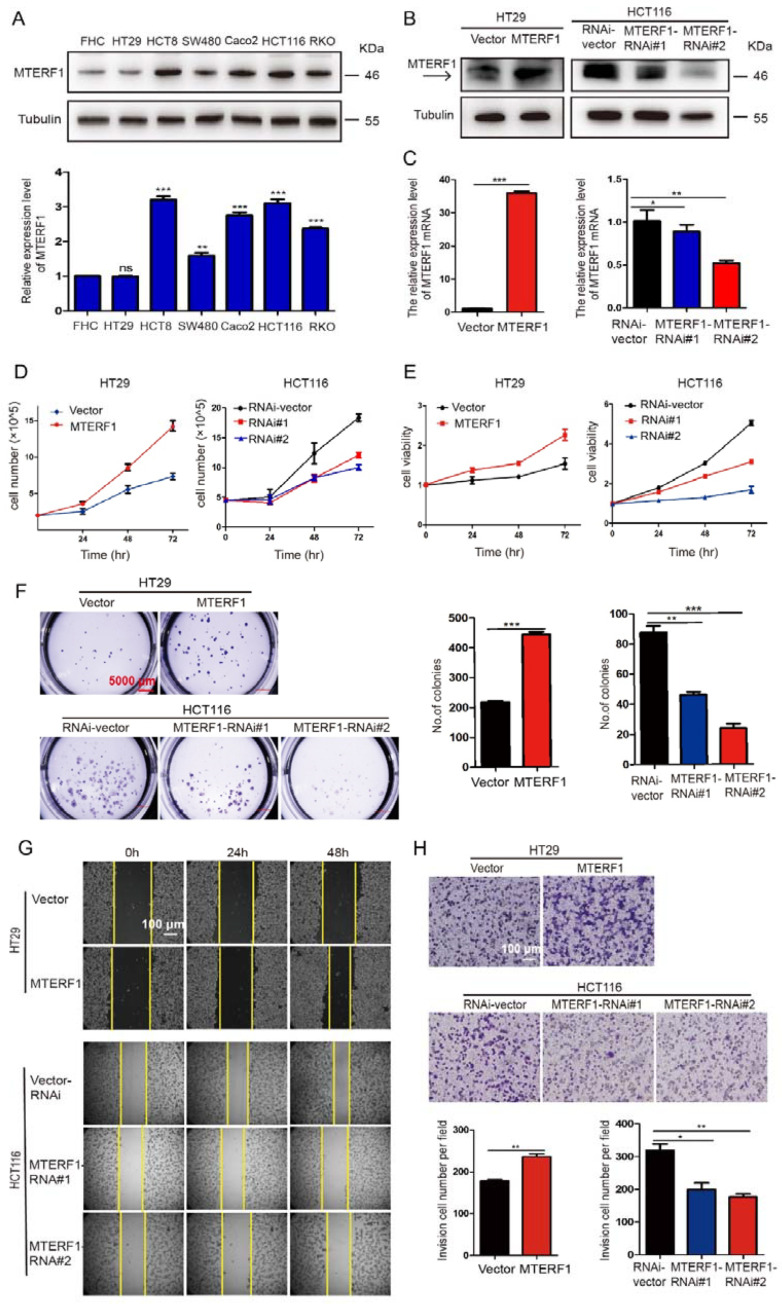
MTERF1 promotes CRC cell proliferation, migration, and invasion. (**A**) Relative MTERF1 protein expression in different colorectal cell lines: normal human colorectal mucosal cell (FHC) and colorectal cancerous cell lines (HT-29, HCT-8, SW480, Caco-2, HCT116, and RKO). (**B**,**C**) Efficiency of MTERF1 overexpression in HT-29 cells and individual MTERF1-shRNA knockdown in HCT116 cells verified by Western blotting (**B**) and real-time PCR (**C**). Vector: pCDH; MTERF1: MTERF1-pCDH. α-Tubulin was used as an internal control, and RNAi-Vector was used as a scramble shRNA control. (**D**) Overexpression of MTERF1 significantly promoted HT-29 cell growth, while knockdown of MTERF1 inhibited HCT116 cell growth. (**E**) Overexpression of MTERF1 significantly enhanced HT-29 cell viability, while knockdown of MTERF1 inhibited HCT116 cell viability. (**F**) Overexpression of MTERF1 promoted HT-29 cell colony formation ability, while knockdown of MTERF1 inhibited HCT116 cell colony formation ability. (**G**) Wound healing assay showed that overexpression of MTERF1 promoted the migration of HT-29 cells, and knockdown of MTERF1 inhibited the migration of HCT116 cells. Scale bar: 100 μm. (**H**) Overexpression of MTERF1 increased HT-29 Transwell cell invasion (48 h), and knockdown of MTERF1 by independent shRNAs decreased HCT116 Transwell cell invasion (48 h). Scale bar: 100 μm. Data are presented as the means ± SEM (n ≥ 3). NS represents no significant difference (*p* > 0.05). * *p*< 0.05, ** *p*< 0.01, *** *p* < 0.001 by two-way ANOVA and two-tailed Student’s *t*-test.

**Figure 3 ijms-23-12354-f003:**
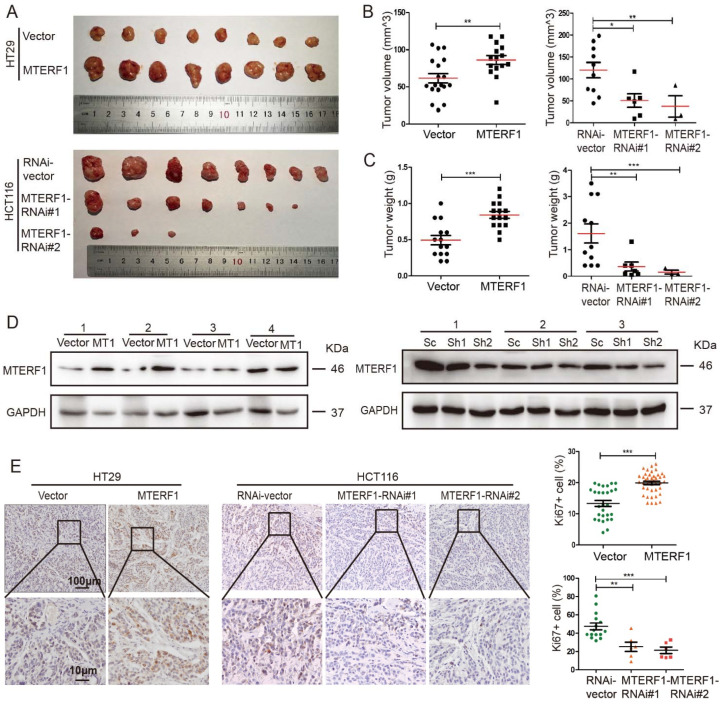
MTERF1 is beneficial to xenograft tumor formation in vivo. (**A**) Upregulation of MTERF1 (MTERF1) promoted xenograft tumor formation, while downregulation of MTERF1 inhibited xenograft tumor formation in vivo. (**B**,**C**) Quantitative analysis of xenograft tumor volume (**B**) and weight (**C**). (**D**) Detection of MTERF1 expression in xenograft tumor samples by Western blotting analysis. V: Vector; MT1: MTERF1; Sc: RNAi-vector; Sh1: MTERF1-RNAi#1; Sh2: MTERF1-RNAi#2. GAPDH was used as an internal control. (**E**) Representative images of Ki67 IHC and quantification data for xenograft tumor sections. Scale bar: 100 μm. Scale bar: 25 μm. Data are presented as the means ± SEM (n ≥ 3). * *p* < 0.05, ** *p* < 0.01, *** *p* < 0.001 by two-tailed Student’s *t*-test.

**Figure 4 ijms-23-12354-f004:**
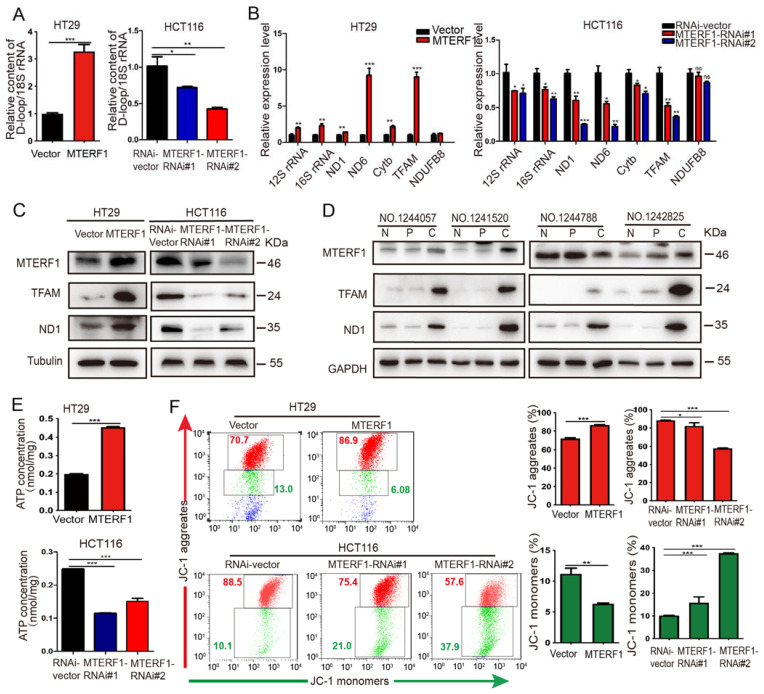
MTERF1 regulates mitochondrial gene expression and OXPHOS in CRC cells. (**A**) Detection of the relative content of D-loop/18S rDNA by qPCR in HT-29 and HCT116 cells upon upregulation and downregulation of MTERF1. (**B**,**C**) qRT–PCR analysis (**B**) and Western blot analysis (**C**) of the expression levels of mitochondrial coding genes and mitochondrial transcription-regulated genes. α-Tubulin was used as an internal control. (**D**) Western blot analysis of the expression levels of MTERF1, TFAM, and ND1 in clinical CRC tissues (**C**), paired paracarcinoma tissues (P), and paired normal tissues (N). GAPDH was used as an internal control. (**E**) Detection of ATP concentration in HT-29 and HCT116 cells. (**F**) Detection of mitochondrial membrane potential (ΔΨm) in HT-29 and HCT116 cells using JC-1 staining. Reduction of ΔΨm prevents the accumulation of JC-1 in the mitochondria and disperses throughout the cells, leading to a shift from red (JC-1 aggregates) to green fluorescence (JC-1 monomers). NS represents no significant difference. * *p* < 0.05, ** *p* < 0.01, *** *p* < 0.001, two-tailed Student’s *t*-test. Data represent the mean ± SD from at least three independent biological replicates.

**Figure 5 ijms-23-12354-f005:**
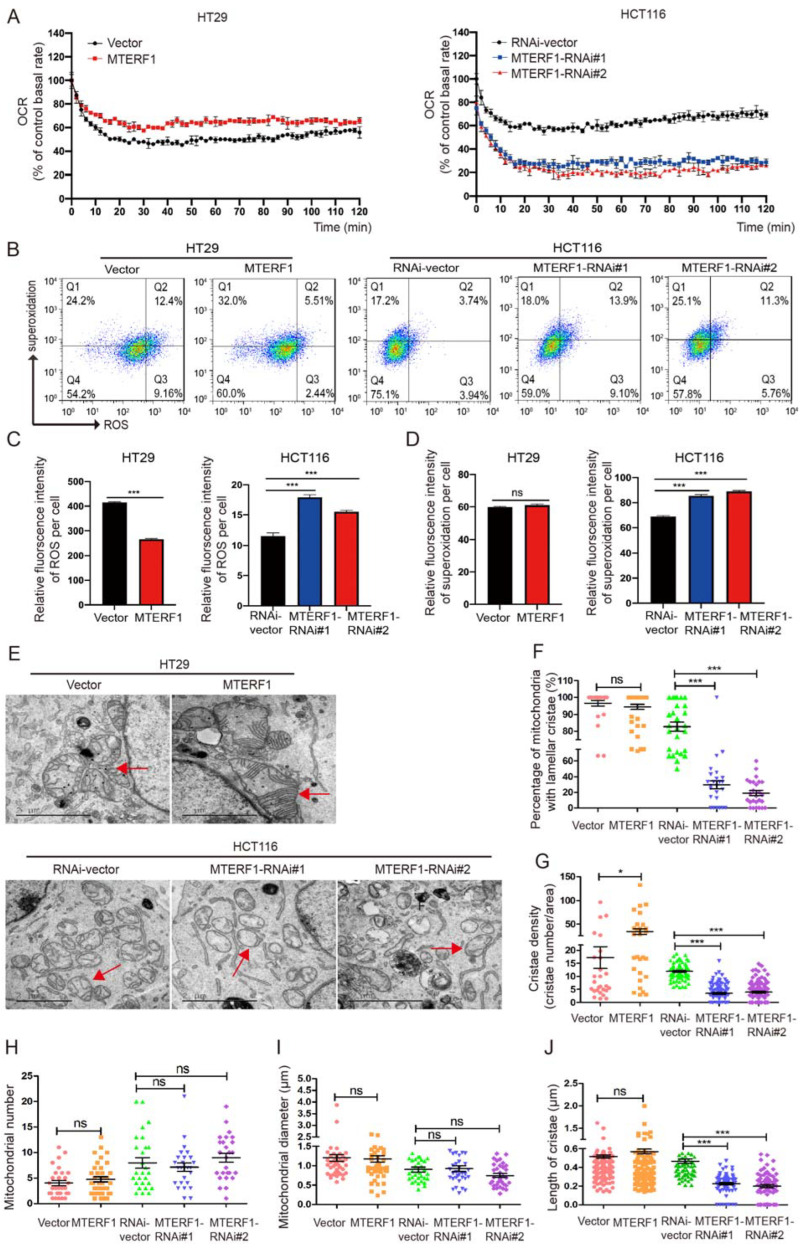
Effect of MTERF1 on the OCR, ROS levels, and mitochondrial inner membrane ultrastructure of colorectal cancer cells. (**A**) OCR detection in CRC cells with MTERF1 overexpression in HT29 cells and knockdown in HCT116 cells every 10 min. (**B**) Flow cytometry detects intracellular ROS and superoxidation levels. (**C**,**D**) Quantitative analysis of the relative fluorescence intensity of ROS (**C**) and superoxidation levels (**D**). (**E**) Representative transmission electron microscopy (TEM) images of mitochondria (red arrowheads) in CRC cells with MTERF1 overexpression in HT29 cells and knockdown in HCT116 cells. Scale bar: 2 μm. (**F**,**J**) Quantification of the mitochondrial ultrastructural profile, including the mitochondria with lamellar cristae percentage (**F**), cristae density (**G**), mitochondrial number (**H**), mitochondrial diameter (**I**), and cristae length (**J**). NS represents no significant difference. * *p* < 0.05, *** *p* < 0.001, two-tailed Student’s *t*-test. Data represent the mean ± SD from at least three independent biological replicates.

**Figure 6 ijms-23-12354-f006:**
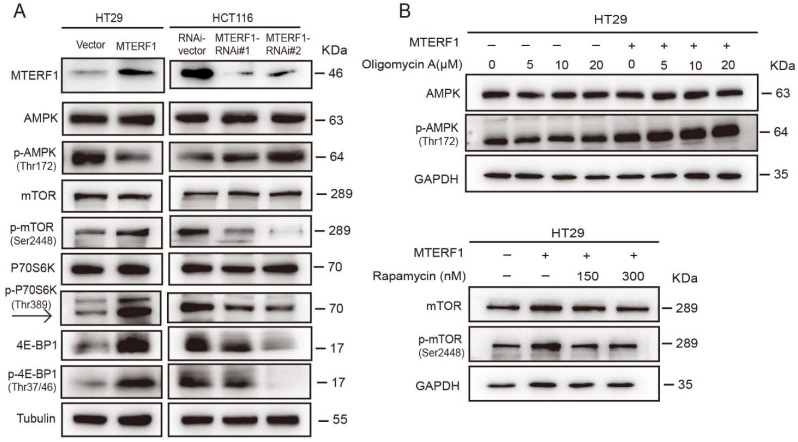
MTERF1 regulates the p-AMPK-mTOR signaling pathway in CRC cells. (**A**) Upregulation of MTERF1 inhibited AMPK phosphorylation and promoted mTOR phosphorylation levels, yet downregulation of MTERF1 had the opposite effect. Data represent the mean ± SD from at least three independent biological replicates. (**B**) HT29 cells overexpressing MTERF1 were treated with 0–20 μM oligomycin A and 150–300 nM rapamycin, and the AMPK/mTOR pathway regulated by MTERF1 was blocked by oligomycin and rapamycin.

**Figure 7 ijms-23-12354-f007:**
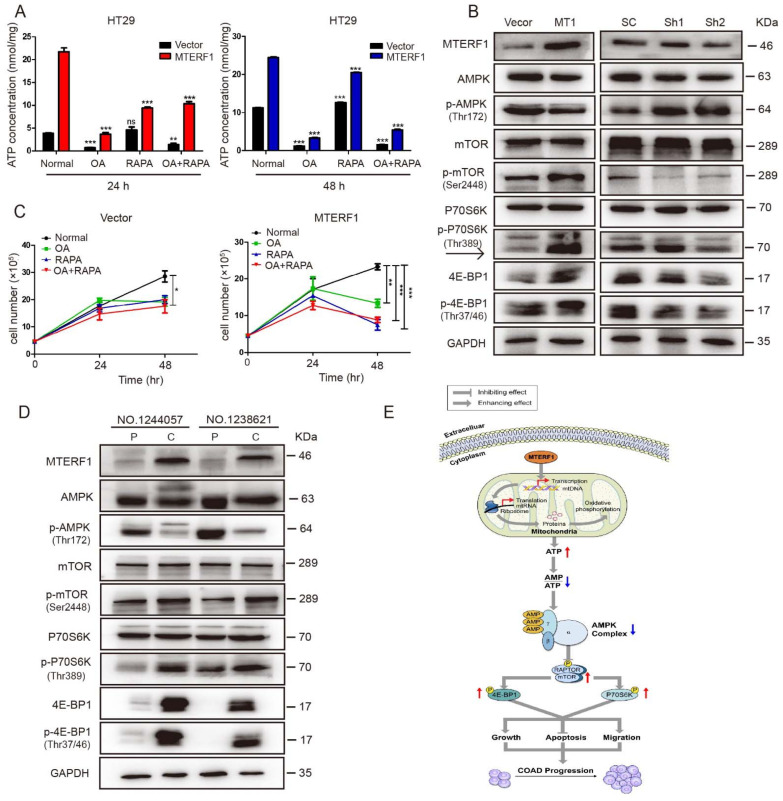
MTERF1 regulates the p-AMPK-mTOR signaling pathway in xenograft tumor tissue and human clinical samples. (**A**,**B**) HT29 cells with MTERF1 overexpression were treated with 5 μM oligomycin A (OA) and 300 nM rapamycin (RAPA) to detect the effect on ATP levels (**A**) and cell proliferation (**B**). (**C**,**D**) Protein expression levels of AMPK/mTOR signaling pathway-related proteins in xenograft tumor tissue (**C**) and human clinical samples (**D**). (**E**) Working model for MTERF1. As proposed in CRC cells, MTERF1 increased mitochondrial replication, transcription, and protein synthesis, which enhanced mitochondrial OXPHOS and ATP levels, thus promoting mTOR phosphorylation levels and finally promoting the occurrence and development of CRC cells. NS represents no significant difference. * *p* < 0.05, ** *p* < 0.01, *** *p* < 0.001, two-tailed Stu-dent’s *t*-test. Data represent the mean ± SD from at least three independent biological replicates.

## Data Availability

Not applicable.

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
