# Peer review of "Modulating p-AMPK/mTOR Pathway of Mitochondrial Dysfunction Caused by MTERF1 Abnormal Expression in Colorectal Cancer Cells"

_ijms, 2022, doi:10.3390/ijms232012354_

Round 1

Reviewer 1 Report

The manuscript "Modulating p-AMPK/mTOR pathway of mitochondrial dysfunction caused by MTERF1 abnormal expression in colorectal cancer cells" is focused on the analysis of an oncogenic role of human mitochondrial transcription termination factor 1 (MTERF1) in colorectal cancer (CRC) progression.

This work identified high expression of MTERF1 in colon cancer tissues compared with normal colorectal tissue by western blotting and immunohistochemistry and found its correlations with four clinicopathological parameters, patient lymph node status, tumor stage, nodal stage, and clinical stage.

 This study also showed that the MTERF1 overexpression in the HT29 cell promoted cell proliferation, migration, invasion, and xenograft tumor formation and growth, whereas MTERF1 knockdown in HCT116 cells appeared to be the opposite phenotype of HT29 cells. Additionally, it was demonstrated that in colorectal cancer cells, MTERF1 can increase different mitochondrial functions, such as mitochondrial DNA replication, transcription, and protein synthesis, increase ATP levels, the mitochondrial crista density, mitochondrial membrane potential, and oxygen consumption rate (OCR) and reduce the ROS production enhancing mitochondrial oxidative phosphorylation (OXPHOS) activity. The functional analysis with oligomycin and rapamycin revealed that the mitochondrial regulator MTERF1 can impact the AMP levels and thereby regulate the AMPK/mTOR signaling pathway in cancerous cell lines in vitro and xenograft tumor tissues.

This manuscript is well written and might interest oncologists and cancer biologists. The significance of the studied issues and the study's novelty are substantiated in the Introduction. Methods and Results are described accurately and sufficiently. In general, the presented results support all the conclusions.

I have the following comments on the manuscript.

1. Clarification is required for all the pictures with Western blots in Figures 1A, 2A, 2B, 3D, 4C, 4D, 6A-C, 7B, and 7D. It is necessary to indicate the size of the detected proteins in each band but not the position between the indicated markers.

2. In the Supplementary files containing complete Western blots, it is necessary to indicate the marks of all western blots' lines and indicate the sizes and positions of protein markers.

Reviewer 2 Report

The manuscript by Liu and coworkers elaborates the role of MTERF1 for the development and progression of colon cancer. The manuscript is interesting, well written and well designed. The provided experiments and described results are accurate for this journal. There are only a few minor points from my side the authors might heed in a revised version: 

Figure 1D: The meaning of this figure is unclear. This figure must be improved or placed in a supplement with bigger size. It also needs a more detailed explanation in the main text.

Figure 4F: This figure is relatively small, too. It should be presented in bigger size, maybe in the supplement.

Section 2.5.: Please also mention in the main text of this section that electron microscopy was applied.

Section 2.6.: 300 µM rapamycin is a quite high concentration for an anticancer compound in cell-based assays. Please explain why such high doses were applied.

Discussion: Please also discuss briefly the relevance of increased ATP levels in terms of cancer drug resistance by fueling membrane ABC-transporters (drug efflux pumps).

Section 4.2.: Please write ´´2´´ in ´´CO2´´ in subscript. The same for ´´O2´´ in section 4.14., and ´´4´´ in ´´OsO4´´ in section 4.15.
